# Effectiveness of implementation strategies for uptake of fall prevention interventions in community care: A systematic review

Siv Linnerud[1], Linda Aimée Hartford Kvæl[1,2], Birgitte Graverholt[3], Maria Bjerk[1,4], Kristin Taraldsen[1], Therese Brovold[1]*

1 Department of Rehabilitation Science and Health Technology, Faculty of Health Sciences, OsloMet - Oslo Metropolitan University, Oslo, Norway, 2 Department of Ageing Research and Housing Studies, Norwegian Social Research (NOVA), OsloMet - Oslo Metropolitan University, Oslo, Norway, 3 Department of Health and Caring Sciences, Faculty of Health and Social Sciences, Western Norway University of Applied Sciences, Bergen, Norway, 4 Division for Health Services, Norwegian Institute of Public Health Oslo, Oslo, Norway

* therbrov@oslomet.no

## Abstract

### Introduction

Falls among older adults are one of the leading causes of morbidity and mortality in this population and pose a significant economic burden on society. Despite substantial evidence supporting effective interventions to prevent falls, their uptake in practice remains limited. While various strategies could enhance implementation, few have been evaluated for their effectiveness in fall prevention. This study aimed to assess the effectiveness of such implementation strategies on implementation and health-related outcomes in community care for fall prevention.

### Method

A systematic search was conducted in MEDLINE (Ovid), CINAHL, EMBASE, PsycINFO, Web of Science, the Cochrane Library, and Google Scholar November 27th 2024. We included studies evaluating an implementation strategy for preventing falls among older adults living in the community. Eligible study designs included randomized controlled trials (RCTs), non-randomized trials, controlled before-and-after studies, interrupted time series, and repeated measures studies. The results were narratively synthesized using vote-counting based on the direction of effect. The risk of bias was assessed using RoB2 and ROBINS-I.

### Results

This review identified 8365 distinct references, leading to the inclusion of four articles derived from three unique studies. Our findings suggest that employing a systematic implementation strategy aimed at training and educating stakeholders can enhance

**Data availability statement:** All relevant data are within the manuscript and its Supporting Information files.

**Funding:** This work was supported by the Research Council of Norway (Grant number 301996).

**Competing interests:** The authors have declared that no competing interests exist.

the number of fall preventive interventions hosted by counties and communities. However, the results did not indicate a clear direction of effect concerning hospital or emergency department admissions.

## Conclusion

Our review underscores a significant gap in research concerning the effectiveness of implementation strategies for integrating fall prevention interventions into practice. Future research should focus on developing and testing the effectiveness of these strategies in preventing falls among community-dwelling older adults.

## Registration

PROSPERO CDR42022233395.

## Introduction

One-third of older adults' experience one or more falls annually, and falls are one of the leading causes of morbidity and mortality among this group [1]. Falls are considered a major health hazard as they can lead to serious injuries, reduce individuals' health, and increase the need for healthcare services [2]. Over the last decade, there has been significant interest in how to prevent falls, and in 2022, the World Falls Guidelines were published [1]. In general, these guidelines recommend an increased focus on fall identification, multifactorial fall risk assessment, and tailored exercises focusing on strength and balance to reduce falls [1]. Despite solid evidence on how to prevent falls [1,3,4], its implementation in practice has been slow and limited [5,6]. Although fall prevention is regarded as an important task in the community setting, implementation of fall prevention guidelines is considered a complex process that requires systematic work of identifying barriers and facilitators within the health care services [7]. In community settings, implementation strategies have often been used unconsciously and unintentionally, through a trial and error method, to improve practice [7]. Barriers to implementation in the community setting are related to the underreporting of falls, lack of knowledge among healthcare providers, interventions' incompatibility with practice, and the absence of financial incentives [6,8].

The implementation of research evidence into practice is considered a complex process [9]. Accumulated insights from implementation science highlight the need for a systematic approach. Such an approach involves identifying local barriers to the use of specific research [9] and addressing them with implementation strategies or methods designed to overcome these barriers, thus tailoring the uptake of evidence [10]. In 2015, a group of implementation experts formulated the Expert Recommendations for Implementing Change (ERIC) taxonomy, which comprises 73 implementation strategies [11]. The ERIC study aimed to establish a common understanding of implementation strategy terms and definitions by gathering input from diverse stakeholders in implementation science and practice. Several implementation strategies have been assessed for their effectiveness. The use of clinical decision support,

practice visits, audit and feedback, employment of local opinion leaders, tailored interventions, and employee training are examples of implementation strategies that have improved healthcare professionals' adherence to guidelines [12–14]. However, to the best of our knowledge, no reviews have assessed the effectiveness of similar implementation strategies for fall prevention in the community setting. Given that implementation is contextual, there is a need to tailor implementation strategies to fall prevention interventions and further explore effective strategies for implementation [15].

As a result, it remains unclear what the effective implementation strategies for preventing falls in the community setting are, posing a challenge for further practice. An overview of effective implementation strategies can provide insights for future research opportunities and inform the design of fall prevention interventions. Thus, the aim of this study was to assess the effectiveness of implementation strategies on implementation and health-related outcomes in community care for fall prevention.

## Method

The protocol for this systematic review was registered with PROSPERO (ID = CRD42022233395). The Preferred Reporting Items for Systematic Reviews and Meta-Analyses (PRISMA) guideline was followed [16] (Additional file 1), complemented by the Synthesis Without Meta-analysis (SWiM) reporting guidelines [17].

### Information source and search strategy

A systematic search was conducted and updated November 27th 2024, using databases MEDLINE (Ovid), CINAHL, EMBASE, PsycINFO, Web of Science, the Cochrane Library, and Google Scholar. The search strategy was developed based on a similar review from 2011 [15], using MeSH terms and keywords for four elements: 1) Older adults, 2) Falls, 3) Prevention, and 4) Implementation. Additionally, we manually screened the reference lists of the included papers. The full search strategy is included in Additional File 2.

### Eligibility criteria

The inclusion criteria are presented in Table 1. We included implementation studies that aimed to test strategies for the uptake of fall prevention interventions. While we anticipated that healthcare professionals would play a crucial role in most implementation strategies aimed at reducing falls, we also included studies where individuals other than healthcare professionals, such as family members or volunteers, played a more active role. For outcomes, we included all measures used to assess the effectiveness of the implementation strategy. Implementation outcomes considered were acceptability, adoption, appropriateness, feasibility, fidelity, penetration, and sustainability. Clinical outcomes considered eligible were falls and fall related consequences such as injuries etc. Conference abstracts and protocols were excluded.

**Table 1. Inclusion criteria for study selection.**

| | |
|---|---|
| **Population** | Healthcare providers or others involved in the implementation strategy targeted at community-dwelling older adults |
| **Intervention** | Implementation studies aiming to test strategies for the uptake of fall prevention intervention |
| **Comparison** | Fall prevention intervention without implementation strategy |
| **Outcome** | Implementation outcomes or clinical outcomes evaluating the implementation strategy |
| **Study design** | Randomized controlled trials, non-randomized trials, controlled before-after studies, interrupted time series, or repeated measures studies |
| **Publication year** | Studies published after 2010 |
| **Language** | English, Norwegian, Swedish, or Danish |

## Study selection

References from the literature searches were imported into EndNote (version 21) and deduplicated. Titles and abstracts were then independently screened by the first author and a research assistant using the systematic review tool, Covidence [18]. In cases of disagreement at this stage, the article was included for full-text screening. Full-text articles were screened independently by pairs of authors (SL, LAHK, BG, KT, and TB), and then compared. Any disagreements were resolved by involving a third author.

## Quality assessment

Risk of bias was independently assessed by pairs of authors (SL, LAHK, BG, and TB), using Version 2 of the Cochrane Risk of Bias Tool for Randomized Trials (RoB2) and the ROBINS-I tool [19]. The overall quality and certainty of evidence (Grading of Recommendations, Assessment, Development, and Evaluations) were not assessed due to the lack of effect sizes and confidence intervals within the results.

## Data extraction

A data extraction form was developed using Microsoft Excel for 365 spreadsheet (Version 2002). We extracted: author, year, country, participant demographics, descriptions of the fall prevention intervention, details of the implementation strategy, outcome measures, and study results. Data were extracted verbatim by the first author and verified by the last. The ERIC compilation of implementation strategies were used to identify and categorize the implementation strategies [11]. Strategies were further clustered as suggested by Waltz et al. [20]. The table below (Table 2) describes the most relevant implementation strategies from the ERIC compilation [11] expected to be found in the included studies.

## Synthesis methods

As we found variation in the interventions, comparators, and outcomes, we synthesized the results using vote-counting based on the direction of effect, as recommended in the Cochrane handbook [21]. Our approach followed the SWiM reporting guidelines [17] and the synthesis was done regardless of statistical significance [21]. We synthesized results for both implementation and clinical outcomes. For each outcome, the effects were categorized as either in favor of the implementation strategy or the control intervention without an implementation strategy. In cases where more than one outcome could be relevant from the study, we chose the outcome that was most relevant to answering our research question. We did not take the measurement timepoint into consideration. No hypothesis test or calculation of confidence interval for the estimate was conducted due to the low numbers in each analysis [21]. Missing outcome data was evaluated in the Risk of Bias assessment, as vote-counting based on the direction of effect does not account for this.

**Table 2. Most relevant implementation strategies from ERIC compilation [11].**

| Expected objectives of implementation strategies | Expected implementation strategies from ERIC compilation |
|---|---|
| Enhancing training and education | • Develop educational materials<br>• Make training dynamic<br>• Use train-the-trainer strategies |
| Fostering leadership and adoption | • Identify and prepare champions<br>• Identify early adopters |
| Securing financial resources | • Access new funding<br>• Fund and contract for the clinical innovation |
| Engaging stakeholders and building consensus | • Conduct educational meetings<br>• Use advisory boards and workgroups |

## Amendments made to the protocol

Some amendments were made to the PROSPERO protocol (ID = CRD42022233395) due to the publication by Vandervelde et al. [15] that explored strategies used to implement multifactorial fall prevention interventions in the community. The review question was refined, and implementation studies were specified as the intervention of interest. Additionally, a restriction was applied to the control groups, emphasizing that the control group had to receive a fall prevention intervention similar to that of the intervention group.

## Results

### Study selection

We identified 8365 unique references eligible for title and abstract screening (see Fig 1). A total of 179 studies were screened in full text, resulting in the inclusion of four articles from three unique studies [22–25].

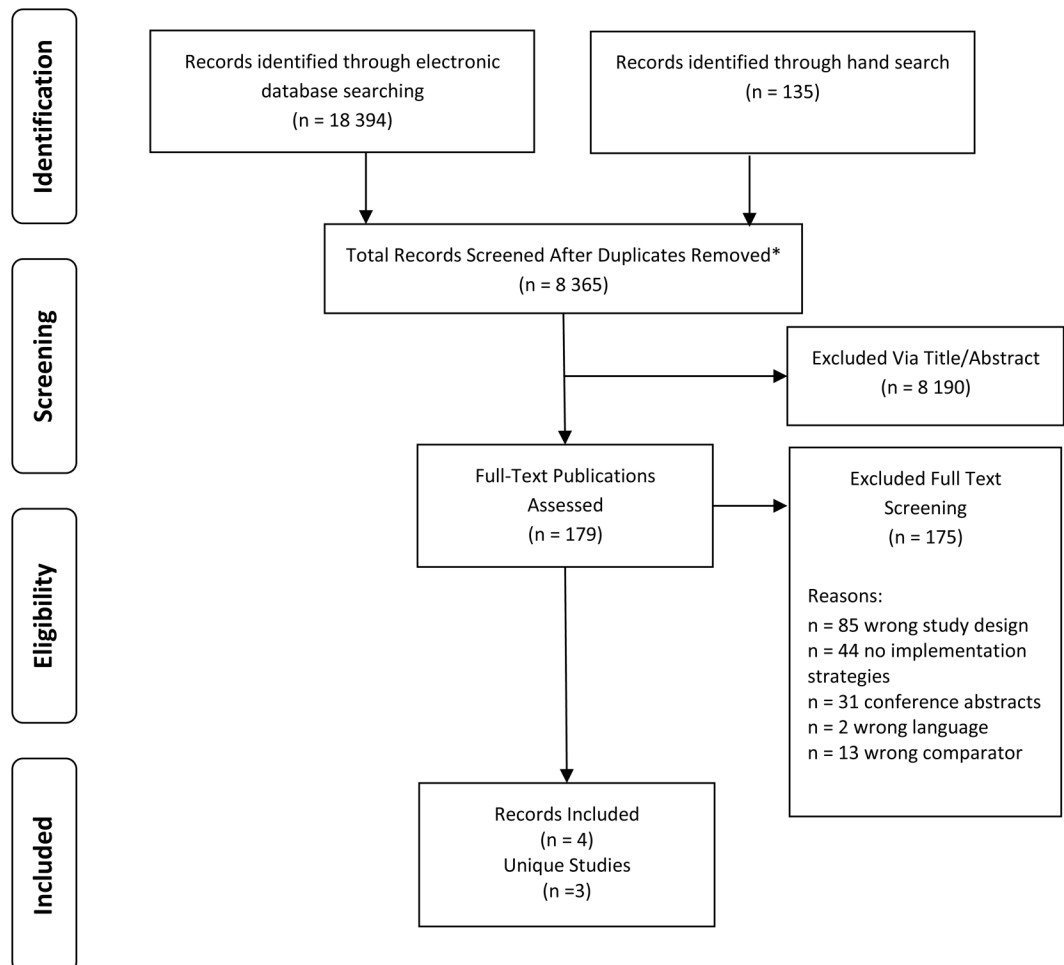

**Fig 1. Flowchart for the inclusion of studies.**

## Study design, setting and fall prevention intervention

Two of the three studies were cluster-randomized at either the community level [23,25] or the county level [22], while the third was a non-randomized controlled trial in which each participating site had both an intervention and a control group [24]. One study was designed as a three-arm randomized trial, testing two implementation strategies [23,25]. All studies were conducted in the United States of America. Study-level characteristics are described in Table 3.

In two out of the three studies [22,23,25], the fall prevention intervention was Stepping On, a multifactorial program that addresses medication review, home modification, strength and balance exercises, community mobility, and vision [22]. Detailed description on how this intervention was conducted was only available for one of the studies, as described by Guse et al. [23] and Peterson et al. [25]. In this study, Stepping On was provided in group settings through 2-hour workshops held weekly over seven weeks. This was followed by a home visit or a phone call at ten weeks and a booster session at three months. In the final study, the fall prevention intervention consisted of screening older adults over 75 years for falls by asking three questions [24]. The results of these three questions were available for both the intervention and the control sites. However, only the intervention site received education on the appropriate response to the results.

## Implementation strategies

The three included studies utilized multifaceted interventions, in which four to six strategies were combined. A total of 15 implementation strategies were described across the three studies. These studies included implementation strategies aimed at training and educating stakeholders, both healthcare providers and older adults. This included making training dynamic [23–25], distributing educational material [24], and conducting educational outreach visits [22]. The implementation strategies are presented in Table 3.

In the study by Ford et al. [22] the implementation intervention involved a change team, including a change leader, formulating and carrying out Plan-Do-Study-Act change cycles over a year. This process was accompanied by coaching, introductory education, and a site visit from the coach, which included an assessment of the context for implementation. The study described by Guse et al. [23] and Peterson et al. [25], had two study arms and one control arm and tested two different implementation strategies within the same study. In the first arm the intervention was called enhanced support, and comprised funding and a training compendium, support consisting of technical assistance and coaching, access to fall data, and an assessment of readiness. n the second arm the implementation intervention, was called standard support, and only comprised funding and a training compendium, [23,25]. Ganz et al. [24] described an implementation intervention that comprised four strategies: training of healthcare providers, use of a decision support system, audit and feedback on healthcare providers' practice, and distribution of patient education handouts.

## Implementation effectiveness

Ganz et al. [24] extracted information on five types of serious injuries from patient medical records as their primary outcome. The clinical outcome in Guse et al. [23] was fall injury morbidity in adults aged 65 years and older, measured by inpatient and emergency department discharge data. This study also included specific implementation outcomes to assess effectiveness of the two implementation interventions, measured as the number of Stepping On workshops, leaders, and participants. Compared to the control arm, only the enhanced support arm was associated with a significantly higher number of workshops, leaders and participants. Additionally, Peterson et al. [25] included the RE-AIM framework (Reach, Effectiveness, Adoption, Implementation, and Maintenance) as a secondary outcome, but they did not report effect estimates between groups for this outcome. Ford et al. [22] used the number of completed workshops and the number of participants enrolled in the workshops as the implementation outcome in their study. They also included the number of falls, and the number of emergency department visits as secondary outcomes but did not report the effect between groups for these outcomes [22].

**Table 3. Characteristics of included studies.**

| First author, year and country | Study design | Sample | Fall prevention intervention | Control group | Implementation intervention | ERIC implementation strategies | Cluster of strategies | Clinical outcome | Implementation outcomes |
|---|---|---|---|---|---|---|---|---|---|
| Ford, 2017 (USA) [22] | CRT | 16 counties | Stepping On | Wait-list control for 1 year, followed by BHAS intervention in year 2 | BHAS intervention | Train-the-trainer | D | Number of emergency department visit and hospitalization. | Number of workshops (Stepping on or Chronic Disease Self-management program), participants enrolled in workshops and participants who completed workshops |
| | | | | | | Conduct educational outreach visits | D | | |
| | | | | | | Recruit, designate and train for leadership | C | | |
| | | | | | | Facilitation | B | | |
| Ganz 2015 (USA) [24] | Non-randomized trial | 4 medical groups in primary care | 3 question falls screening | No implementation strategy | ACOVE prime | Audit and provide feedback | A | Five types of serious injuries: hip fractures, other non-vertebral fractures, inpatient head trauma, joint dislocation, and health care claims where and external cause of injury | |
| Guse, 2015 (USA) [23] and Peterson 2015 (USA) [25] | CRT | 20 communities | Stepping On | State infrastructure support for Stepping On | Enhanced support system | Distribute educational materials | D | Number of unintentional falls in adults aged 65 years and older, as measured by inpatient and emergency department discharge data | Number of workshops (Stepping On) and participants |
| | | | | | | Make training dynamic | D | | |
| | | | | | | Remind clinicians | E | | |
| | | | | | | Facilitate relay of clinical data to providers | E | | |
| | | | | | | Provide local technical assistance | B | | |
| | | | | | | Use implementation advisor | C | | |
| | | | | | | Make training dynamic | D | | |
| | | | | | | Distribute educational materials | D | | |

CRT = Cluster Randomized Trial, BHAS: Bringing Healthy Aging to Scale, ACOVE prime: A multi component quality improvement program, A: Use evaluate and iterative strategies, B: Provide interactive assistance, C: Develop stakeholder interrelationships, D: Train and educate stakeholders, E: Support clinicians, F: Utilize financial strategies.

The vote-counting exercise of implementation outcomes revealed that two studies reported on the number of fall pre-vention interventions held, with both reporting results in favor of the implementation strategy. Both studies incorporated training and education of stakeholders as part of the implementation strategy. This suggests that using training and educa-tion of stakeholders in the implementation strategy resulted in a higher number of fall prevention interventions being held in the counties and thus more older adults being offered the fall prevention intervention [22,23].

The vote-counting exercise for clinical outcomes revealed that two studies reported on fall injuries resulting in emer-gency department or hospital admission. The results for this vote-counting were divergent, with one study favoring the implementation strategy, where using the implementation strategy resulted in fewer fall injuries leading to admission [23]. While the other study favored the control group, where using an implementation strategy resulted in a higher number of fall injuries leading to admission [24]. Both studies used an implementation strategy incorporating training and education of stakeholders. The results of the vote-counting analysis are presented in Table 4. Additionally, Ford et al. [22] reported results that were not part of the vote-counting analysis, as only one result from each study can be included. They indi-cated a significant increase in the number of interventions completed compared to the control group (7.62, p = .05).

### Risk of bias

The risk of bias assessment revealed an overall high risk of bias for three of the outcomes included in the vote-counting analysis, whereas the remaining outcomes were judged to have some concerns. The high risk of bias was related to inad-equate information about the timing of identification or recruitment of the sample, or due to confounding. The outcomes judged to have some concerns were related to a lack of information on deviations from the intended intervention. The risk of bias assessment is presented in Table 5.

**Table 4. Vote counting results direction of effects on implementation outcomes and clinical outcomes.**

| Study (year) | Single strategies of the implementation strategy | Direction of effects | | Results |
|---|---|---|---|---|
| | | Favors Impl. strategy | Favors control | |
| **Implementation outcomes: Number of fall prevention interventions held** | | | | |
| Ford (2017) | Train-the-trainer<br>Conduct educational outreach visits<br>Recruit, designate and train for leadership<br>Facilitation | X | | Number of workshops held/implemented in intervention group (mean 1.38) versus in control group (mean 0.50), p = 0.65 |
| Guse (2015) | Access new funding<br>Assess for readiness and identify barriers and facilitators<br>Conduct local needs assessment<br>Facilitate relay of clinical data to providers<br>Provide local technical assistance<br>Use implementation advisor<br>Make training dynamic<br>Distribute educational materials | X | | Numbers of workshops held combined for the four analytic yeas IRR = 1.62; 95%, CI 1.05, 2.5 |
| **Clinical outcomes: Falls injuries resulting in emergency department or hospital admission** | | | | |
| Ganz (2015) | Audit and provide feedback<br>Distribute educational materials<br>Make training dynamic<br>Remind clinicians | | X | Fall related injuries episodes between interven-tion group and control group during the interven-tion period IRR 1.27; 95% CI 0.93–1.73) |
| Guse (2015) | Access new funding<br>Assess for readiness and identify barriers and facilitators<br>Conduct local needs assessment<br>Facilitate relay of clinical data to providers<br>Provide local technical assistance<br>Use implementation advisor<br>Make training dynamic<br>Distribute educational materials | X | | Number or falls measured as inpatient and emer-gency department discharges from falls from baseline to follow-up within intervention group versus control group IRR 0.918 (0.876, 0.963) |

**Table 5. Results from the risk of bias assessment of outcomes included in the vote-counting analysis.**

| Study | Outcome | D1 | D2 | D3 | D4 | D5 | D6 | D7 | Overall |
|-------|---------|----|----|----|----|----|----|----|---------|
| Ford et al. (2017) | Number of completed workshops, participants enrolled and completed | * | + | − | ! | + | + | ! | − |
| Guse et al. (2015) | Number of workshops, leaders and participants | * | ! | − | ! | + | + | ! | − |
| Guse et al. (2015) | Number of unintentional falls | * | + | + | ! | + | + | + | ! |
| Ganz et al. (2015 | Fall related injuries | − | + | + | + | ! | + | ? | − |

| Domains: | |
|---|---|
| D1: Bias due to confounding<br>D2: Bias in the randomization process/selection of participants into the study<br>D3: Bias in timing of identification or recruitment/classification of interventions<br>D4: Bias due to deviations from intended interventions<br>D5: Bias due to missing outcomes/missing data<br>D6: Bias in measurement of outcomes<br>D7: Bias in selection of the reported result | + Low risk<br>! Some concern/Moderate<br>− High risk/Serious<br>* Not assessed in ROB2 |

## Discussion

This systematic review synthesized results from four articles, reporting three unique studies, that assessed the effectiveness of implementation strategies used to enhance fall prevention interventions for community-dwelling older adults. The studies employed various multifaceted implementation strategies, whereof all contained single strategies aimed at training and educating healthcare providers or service users. The vote-counting analysis demonstrated that employing an implementation strategy can be useful in increasing the number of fall prevention workshops held by counties and communities. However, the use of implementation strategies did not consistently reduce the number of falls leading to referral to acute care.

Surprisingly, only three unique studies met our eligibility criteria. The reasons for the low number of relevant studies are not obvious. One plausible reason could be that there has been lacking interest for implementation and challenging to publish studies with such focus. Additionally, it could be related to less funding for implementation science, which results in less expensive study designs. Another reason may be lack of implementation scientists in research teams. Other systematic reviews exploring implementation strategies have been less rigid regarding the designs included. For instance, Vandervelde et al. [15] imposed no restriction on study design when providing an overview of strategies used to implement multifactorial fall prevention interventions in the community. They included qualitative studies and studies conducted in various settings. Additionally, they utilized a slightly different definition of implementation strategies, describing them as methods or techniques aimed at enhancing the adoption of a "clinical intervention." Still, they included three RCTs and three non-RCTs out of the 18 studies included [15]. However, the systematic review by Vandervelde et al. [15] aimed at summarize implementation strategies in multifactorial fall prevention interventions and had other eligibility criteria than used in this current systematic review. Similar results were seen in the review of Goodwin et al. [26] on the effectiveness of methods to implement falls prevention programs among community dwelling older adults. They neither limited the eligible criteria for study design, and only one of the six included studies had a controlled design. Despite this, randomized controlled designs are the most robust design to compare implementation strategies and evaluate effectiveness of implementation strategies [27].

The implementation strategies described in this review represented clusters of single strategies, where all of them aimed at training and educating stakeholders (healthcare providers or older adults) to enhance the implementation of fall prevention interventions. This was not surprising, as these strategies are suggested to enhance competence levels of the innovation, willingness to change, and create motivation among stakeholders [28]. The use of strategies to train and educate stakeholders was also identified as key in multifactorial fall prevention interventions, and more specifically, active learning, in the systematic review by Vandervelde et al. [15]. The use of education as part of the implementation strategy,

to enhance competence is often employed in other clinical areas as well, such as the implementation of practice guidelines [14] or guidelines in nursing [29].

The vote-counting analysis for clinical outcomes yielded one study in favor of the implementation strategy and one in favor of the control, thereby not providing a clear direction of overall effect for the outcome. With so few studies reporting on clinical outcomes, it is difficult to determine if more studies would have provided a different result. It has been suggested to not employ clinical outcomes as measures of effect of implementation strategies, as they make it harder to distinguish the effects of the fall prevention intervention from the effects of the implementation strategy [30]. The progress of employing implementation outcomes has been criticized for being slow [31]. This also seems to be the case for implementation strategies for fall prevention, as only two of the included studies in our review reported effect estimates on implementation outcomes.

The included studies did not describe in detail their implementation strategy and lacked sufficient information for comparison. This finding aligns with the systematic review of Vandervelde et al. [15], indicating poor reporting practice when reporting guidelines have been available since 2013 [32]. Consequently, when implementation studies are included in systematic reviews the poor reporting also influenced the risk of bias assessment. The outcomes in our review were all assessed as either some concern or with high risk of bias.

### Strengths and limitations

The strength of this review lies in the systematic nature of the review process and the transparency in reporting amendments and the use of PRISMA and SWiM reporting guidelines. The search strategy included going through the literature lists of relevant publications and systematic reviews identified in the search. Nevertheless, we did not search for grey literature or perform forward snowballing. Therefore, it is possible that we missed relevant studies. The included studies also demonstrated poor reporting on implementation strategies, thereby challenging the comparisons of implementation strategies. To ensure transparency in the review process, we registered the protocol in PROSPERO. However, the changes to the protocol may have introduced bias. These changes included specification of the aim, type of intervention, and control/comparison condition.

### Implications for practice and research

This study does not provide sufficient evidence to determine which implementation strategies are most effective to employ when implementing fall prevention interventions in practice. For the implementation of such interventions, the use of a multifaceted implementation strategy should be considered to increase the success of these strategies and the uptake of evidence. Future research should consider providing thorough descriptions of implementation strategies and employing robust designs to further explore the effectiveness of these strategies and provide results on which ones to use.

## Conclusion

This systematic review highlights a gap in research on the effectiveness of implementation strategies for the uptake of fall prevention interventions in practice. Employing an implementation strategy aimed at training and education stakeholders can be useful for increasing engagement in conducting fall prevention intervention. Considering the substantial evidence supporting effective interventions for fall prevention, it is critical to prioritize implementation studies in the field and secure financial support for implementation research. Researchers should strive to provide detailed descriptions of implementation strategies in accordance with reporting guidelines. There is a need for future studies with robust designs to determine the effectiveness of implementation strategies in fall prevention among community-dwelling older adults.

## Supporting information

**S1 Fig. Prisma checklist.** Additional file 1: Prisma checklist.
(DOCX)

**S2 Table. Search strategy.** Additional file 2: Search strategy.
(DOCX)

**S3 Table. List of all included references.** Additional file 3: List of all included references.
(XLSX)

## Acknowledgments

The authors are thankful to the research librarian Marlene Wøhlk Gundersen who assisted in the literature searches. We also want to acknowledge the research assistant, Thale Kristin Holtan, who participated in the initial screening process.

## Author contributions

**Conceptualization:** Siv Linnerud, Linda Aimée Hartford Kvæl, Birgitte Graverholt, Therese Brovold.

**Data curation:** Linda Aimée Hartford Kvæl, Birgitte Graverholt, Maria Bjerk, Therese Brovold.

**Formal analysis:** Siv Linnerud, Linda Aimée Hartford Kvæl, Kristin Taraldsen, Therese Brovold.

**Methodology:** Siv Linnerud, Linda Aimée Hartford Kvæl, Birgitte Graverholt, Maria Bjerk, Kristin Taraldsen, Therese Brovold.

**Project administration:** Siv Linnerud.

**Supervision:** Birgitte Graverholt, Therese Brovold.

**Writing – original draft:** Siv Linnerud.

**Writing – review & editing:** Siv Linnerud, Linda Aimée Hartford Kvæl, Birgitte Graverholt, Maria Bjerk, Kristin Taraldsen, Therese Brovold.

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
