## [Decision Letter · Decision Letter 0]

Dear Dr. Linnerud,

Thank you for submitting your manuscript to PLOS ONE. After careful consideration, we feel that it has merit but does not fully meet PLOS ONE’s publication criteria as it currently stands. Therefore, we invite you to submit a revised version of the manuscript that addresses the points raised during the review process.

The authors of this systematic review aimed to assess the effectiveness of implementation strategies for uptake of fall prevention interventions in community settings. The review is relevant however, my major concern is that the search is over a year old. The search date is a concern for two reasons: 

firstly, as the authors have themsleves highlighted that the review by Vandervelde et al is of a similar scope and is very recent;

secondly, there are only a few studies included in the review and any recent study might have major implications on the results of the review.

I would suggest that if the authors could update the search, the results would be more recent and relevant.

We look forward to receiving your revised manuscript.

Kind regards,

Rehana Abdus Salam

Academic Editor

PLOS ONE

Journal Requirements:

2. Thank you for stating the following financial disclosure: [This work was supported by the Research Council of Norway (Grant number

301996).]. Please state what role the funders took in the study. If the funders had no role, please state: "The funders had no role in study design, data collection and analysis, decision to publish, or preparation of the manuscript." If this statement is not correct you must amend it as needed. Please include this amended Role of Funder statement in your cover letter; we will change the online submission form on your behalf.

3. In this instance it seems there may be acceptable restrictions in place that prevent the public sharing of your minimal data. However, in line with our goal of ensuring long-term data availability to all interested researchers, PLOS’ Data Policy states that authors cannot be the sole named individuals responsible for ensuring data access (http://journals.plos.org/plosone/s/data-availability#loc-acceptable-data-sharing-methods). Data requests to a non-author institutional point of contact, such as a data access or ethics committee, helps guarantee long term stability and availability of data. Providing interested researchers with a durable point of contact ensures data will be accessible even if an author changes email addresses, institutions, or becomes unavailable to answer requests. Before we proceed with your manuscript, please also provide non-author contact information (phone/email/hyperlink) for a data access committee, ethics committee, or other institutional body to which data requests may be sent. If no institutional body is available to respond to requests for your minimal data, please consider if there any institutional representatives who did not collaborate in the study, and are not listed as authors on the manuscript, who would be able to hold the data and respond to external requests for data access? If so, please provide their contact information (i.e., email address). Please also provide details on how you will ensure persistent or long-term data storage and availability.

5. As required by our policy on Data Availability, please ensure your manuscript or supplementary information includes the following:

Additional Editor Comments:

The authors of this systematic review aimed to assess the effectiveness of implementation strategies for uptake of fall prevention interventions in community settings. The review is relevant however, my major concern is that the search is over a year old. The serach date is a concern for two reasons:

firstly, as the authors have themsleves highlighted that the review by Vandervelde et al is of a similar scope and is very recent;

secondly, there are only a few studies included in the review and any recent study might have major implications on the results of the review.

I would suggest that if the authors could update the search, the results would be more recent and relevant.

1. Line 130: "In case of disagreement...."

2. The authors have used the terms 'recommendations' and 'interventions interchangeably and I would suggest consistency for readers.

3. This statement in the Disucssion section is not clear: "the use of an implementation strategy did not provide unambiguous effects for falls leading to referral to acute care."

4. I think that it is important to specify how the authors eligibiltiy criteria differed from the review by Vandervelde et al in the disucsison sestion where they state: "Vandervelde et al. (14) aimed at summarize implementation strategies in multifactorial fall prevention interventions and had other eligibility criteria than used in this current systematic review." especially when they have inlcuded three RCTs and three non_RCTs that do not overlap with the included studies in this review.

4. Line 319: Change Prospero to PROSPERO

Reviewers' comments:

Reviewer's Responses to Questions

**Comments to the Author**

1. Is the manuscript technically sound, and do the data support the conclusions?

Reviewer #1: Yes

2. Has the statistical analysis been performed appropriately and rigorously?

Reviewer #1: N/A

3. Have the authors made all data underlying the findings in their manuscript fully available?

Reviewer #1: Yes

4. Is the manuscript presented in an intelligible fashion and written in standard English?

Reviewer #1: Yes

Reviewer #1: Thank you for giving me the opportunity to review this manuscript. This is a very important area, which requires commitment from stakeholders to deliver effectively, and can be challenging in local contexts. Evidence of effective implementation strategies are needed.

Introduction

The literature to date is clearly presented, with good level of detail on other reviews in the area.

Methods

Supported with appropriate guidelines and pre-registered.

Line 143: was the implementation strategy extracted verbatim? see comment on Table 1 below.

Lines 202 – 214: I find this section difficult to follow. Suggest “the strategy used by…” rather than “ the first strategy”, or add some more context. Were the studies comparing the effects between the standard and enhanced support? In line with Table 1, suggest describing Ford et al’s study before the others.

Table 1: A number of changes may help readability here. Can I suggest to add reference numbers to the studies listed; I initially read Guse 2015 and Paterson 2015 as one report – two citation numbers would help here. Add “1st” to Author. Explain BHAS and ALCOVE abbreviations. Is “cluster of single strategies” described in the text? What guided your categorization of these into A-F? I would be inclined to add more detail to the interventions in the table.

Results

Results are clearly presented and demonstrate the challenge to coming to a conclusive answer.

Strength, Limitations and Conclusions are clear and well presented.

Presentation:

There is a different font used in Table 1 for “comparison”. Line 130 [I]n cases of disagreement….

Figure 1 was not clear in the PDF document.

**Do you want your identity to be public for this peer review?** For information about this choice, including consent withdrawal, please see our Privacy Policy

Reviewer #1: No

---

## [Author Response · Author response to Decision Letter 1]

16 Jan 2025

Dear Dr Rehana Abdus Salam

Thank you for considering our manuscript and allowing us to resubmit it. You and the reviewer have done an excellent job in evaluating our work, and we acknowledge the importance of the issues regarding the date of the search. We have updated the search and screened and reviewed the eligibility of 1196 new references. The update of the search has been included in the changes to the manuscript and in Figure 1.

For the additional comments, we have provided a “Point-by-Point” response. We have included a numbered quotation from you and reviewer 1, followed by our response, labeled “AR” for “Author Response” along with a corresponding numerical identifier (e.g., AR3). The responses include references to line numbers within the manuscript where changes have been made.

We believe we have adequately addressed the concerns and suggestions raised by the reviewer. The revised manuscript, including all changes from the original submission, is marked using track changes.

On behalf of all the authors, I would like to express our gratitude for considering this article, and we look forward to your response.

Sincerely,

Siv Linnerud

PhD Student

OsloMet – Oslo Metropolitan University

---

## [Editor Report · Decision Letter 1]

Dear Dr. Linnerud,

Thank you for submitting your manuscript to PLOS ONE. After careful consideration, we feel that it has merit but does not fully meet PLOS ONE’s publication criteria as it currently stands. Therefore, we invite you to submit a revised version of the manuscript that addresses the points raised during the review process.

I would like to thank the authors for revising the draft. However, the manuscript still lacks clarity with regards to the eligibility criteria. This could be due to the fact that the authors had to ammend their protcol due to a recent publication in the same domain. However, the methodology appears to lack the rigor and I have highlighted quite a few major revisions required in order to consider the draft again for publication.

We look forward to receiving your revised manuscript.

Kind regards,

Rehana Abdus Salam

Academic Editor

PLOS ONE

**Additional Editor Comments:**

I would like to thank the authors for revising the draft. However, the manuscript still lacks clarity with regards to the eligibility criteria. This could be due to the fact that the authors had to ammend their protcol due to a recent publication in the same domain. However, the methodology appears to lack the rigor and I have highlighted quite a few major revisions required in order to consider the draft again for publication:

ABSTRACT:

1. Use of term 'clinical' in relation to the interventions is a bit confusing considering that the review includes studies conducted in community care rather than clinical or hospital settings. Use of the term 'clinical' in relation to the outcomes is fine since the those outcomes refer to ER admissions or hospitalisations. I would suggest if the authors could reconsider using the term 'clinical' in relation to the interventions in the abstract as well as elsewhere in the draft.

INTRODUCTION:

The introduction is lengthy and fails to make a good enough case for evaluating implementation strategies for interventions preventing falls in community settings. I have made a few suggestions below to improve that:

1. In line with the first comment under 'ABSTRACT', there is a lot of dicussion around uptake of implementation strategies in clinical practice. However, the authors do not make a good enough case for current practice and uptake in commuity settings.

2. The third paragraph in the 'Introduction' section appears to be more of a 'Discussion' around similarities and differences between this review and other existing reviews. I would suggest that the authors either move it to discussion or completely remove it if the same references are already being discussed in the 'Discussion' section.

3. Paragraph 4 seems repetitive of what has already been established in the end of paragraph 2. For e.g. "When implementing clinical guidelines in general, previous systematic reviews have identified clinical decision support, practice visits, audit and feedback, use of local opinion leaders, tailored interventions, and training as effective strategies. In addition, commonly used strategies for fall prevention include the involvement of stakeholders and tailoring. However, none of these reviews have evaluated the effectiveness of these implementation strategies specifically for fall prevention." can be deleted.

METHODS:

1. Please state all the outcomes considered under 'Implementation Outcomes' and 'Clinical Outcomes' separately. Later in the results section, authors mention the Proctors’ taxonomy of implementation outcomes. I would suggest clarifying in the methodology if these were considered or removing them from the Results section.

2. Considering that the authors have focused on the effectiveness of implementation strategies rather than interventions, I feel that it merits some elaboration in the methods section. For e.g. under the section on 'data extraction' when authors state 'details of implementation strategy' some details can be added on what implementation strategies were anticipated? which implementation strategies did author consider eligible for inclusion? Moreover, the statements like 'single strategies within each identified implementation strategy' could also be very confusing to readers who are not implementation scientists and hence some description of ERIC implementation strategies would be helpful to provide some insight on what are implementation strategies and what was eligible for inclusion in this review. The authors can also add a supplementary table for further reading here.

RESULTS:

1. Not sure why the authors have referred to the Proctors’ taxonomy of implementation outcomes under the section 'Implementation strategy' as it was not discussed in the methods section (as I have highlighted earlier). I would suggest that the authors either add the details in the methods section if any data was extracted based on Proctors' implementation outcomes or delete this statement from the Results section.

2. If the implementation strategies are highlighted earlier in the methods section as suggested, it would be easier for the readers to comprehend the results detailed under the section reporting on 'implementation strategies'.

3. Again, the statement "The second implementation strategy, known as the standard support, comprised funding and a training compendium, similar to the enhanced support (24, 28). For the purpose of this review, only the enhanced strategy has been included." does not make sense since the eligilibity of implementation strategies were not detailed in the methods section. I would suggest that the authors clearly define the implementation strategies considered for inclusion in the 'methods' section in order to guide the 'Results.

4. I would suggest that the authors separately report the implementation outcomes and the clinical outcomes in the text as well as they have reported in the table.

---

## [Author Response · Author response to Decision Letter 2]

12 Mar 2025

Response to editor comments

I would like to thank the authors for revising the draft. However, the manuscript still lacks clarity with regards to the eligibility criteria. This could be due to the fact that the authors had to ammend their protcol due to a recent publication in the same domain. However, the methodology appears to lack the rigor and I have highlighted quite a few major revisions required in order to consider the draft again for publication:

ABSTRACT:

1. Use of term 'clinical' in relation to the interventions is a bit confusing considering that the review includes studies conducted in community care rather than clinical or hospital settings. Use of the term 'clinical' in relation to the outcomes is fine since the those outcomes refer to ER admissions or hospitalisations. I would suggest if the authors could reconsider using the term 'clinical' in relation to the interventions in the abstract as well as elsewhere in the draft.

AR1: Thank you for this important comment. We have considered the use of the term “clinical” in relation to interventions and have addressed this by removing the wording from the abstract and manuscript.

INTRODUCTION:

The introduction is lengthy and fails to make a good enough case for evaluating implementation strategies for interventions preventing falls in community settings. I have made a few suggestions below to improve that:

1. In line with the first comment under 'ABSTRACT', there is a lot of dicussion around uptake of implementation strategies in clinical practice. However, the authors do not make a good enough case for current practice and uptake in commuity settings.

AR1: Thank you for addressing this lack in the introduction. We have addressed current practice and uptake in community setting on page 4, lines 57-62.

2. The third paragraph in the 'Introduction' section appears to be more of a 'Discussion' around similarities and differences between this review and other existing reviews. I would suggest that the authors either move it to discussion or completely remove it if the same references are already being discussed in the 'Discussion' section.

AR2: Thank you for pointing this out. We have suggested removing the section from the introduction. Please see page 5, lines 82-92.

3. Paragraph 4 seems repetitive of what has already been established in the end of paragraph 2. For e.g. "When implementing clinical guidelines in general, previous systematic reviews have identified clinical decision support, practice visits, audit and feedback, use of local opinion leaders, tailored interventions, and training as effective strategies. In addition, commonly used strategies for fall prevention include the involvement of stakeholders and tailoring. However, none of these reviews have evaluated the effectiveness of these implementation strategies specifically for fall prevention." can be deleted.

AR3: Thank you for pointing this out. We have suggested removing the suggested section from the introduction. Please see page 5, lines 93-98.

METHODS:

1. Please state all the outcomes considered under 'Implementation Outcomes' and 'Clinical Outcomes' separately. Later in the results section, authors mention the Proctors’ taxonomy of implementation outcomes. I would suggest clarifying in the methodology if these were considered or removing them from the Results section.

AR1: Thank you. We have added all the outcomes considered in the text describing eligibility criteria. Please see page 7, lines 125-129. Additionally, we have removed the mention of Proctors taxonomy of implementation outcomes from the Results section, please see page 12, lines 209-210.

2. Considering that the authors have focused on the effectiveness of implementation strategies rather than interventions, I feel that it merits some elaboration in the methods section. For e.g. under the section on 'data extraction' when authors state 'details of implementation strategy' some details can be added on what implementation strategies were anticipated? which implementation strategies did author consider eligible for inclusion? Moreover, the statements like 'single strategies within each identified implementation strategy' could also be very confusing to readers who are not implementation scientists and hence some description of ERIC implementation strategies would be helpful to provide some insight on what are implementation strategies and what was eligible for inclusion in this review. The authors can also add a supplementary table for further reading here.

AR2: We appreciate your feedback regarding the limitations in our methods section. We have expanded the 'Data Extraction' section to include detailed information on expected implementation strategies, clarifying our inclusion criteria. A table outlining these strategies has been added. See changes on page 8, lines 150-158. Further, to enhance clarity, we have differentiated between single implementation strategies, henceforth referred to as implementation strategies, and broader implementation interventions that encompass a combination of strategies throughout the manuscript.

RESULTS:

1. Not sure why the authors have referred to the Proctors’ taxonomy of implementation outcomes under the section 'Implementation strategy' as it was not discussed in the methods section (as I have highlighted earlier). I would suggest that the authors either add the details in the methods section if any data was extracted based on Proctors' implementation outcomes or delete this statement from the Results section.

AR1: Thank you for addressing this aspect. We have suggested removing the sentence referring to Proctors taxonomy of implementation outcomes. Please see page 12, lines 209-210.

2. If the implementation strategies are highlighted earlier in the methods section as suggested, it would be easier for the readers to comprehend the results detailed under the section reporting on 'implementation strategies'.

AR2: Thank you for highlighting this point. We have expanded the description of implementation strategies in the methodology section, as detailed in the AR2 under the 'Methods' subheading.

3. Again, the statement "The second implementation strategy, known as the standard support, comprised funding and a training compendium, similar to the enhanced support (24, 28). For the purpose of this review, only the enhanced strategy has been included." does not make sense since the eligilibity of implementation strategies were not detailed in the methods section. I would suggest that the authors clearly define the implementation strategies considered for inclusion in the 'methods' section in order to guide the 'Results.

AR3:Thank you for pointing this out. We now see that the way it is written could be misunderstood. We did not select implementation strategies based on predefined definitions; rather we included all the strategies from the included studies. We have rewritten the information from Guse and in the manuscript, and we hope it is clearer now. See page 13 Line 238-240.

4. I would suggest that the authors separately report the implementation outcomes and the clinical outcomes in the text as well as they have reported in the table.

AR4: Thank you for pointing this out. We believe the implementation and clinical outcomes are reported separately in the text. To further clarify, we have now created distinct paragraphs for each category. This change is reflected on page 13 and 14, lines 246-247.

---

## [Editor Report · Decision Letter 2]

Effectiveness of implementation strategies for uptake of fall prevention interventions in community care: A systematic review

PONE-D-24-39287R2

Dear Dr. Linnerud,

We’re pleased to inform you that your manuscript has been judged scientifically suitable for publication and will be formally accepted for publication once it meets all outstanding technical requirements.

Kind regards,

Rehana Abdus Salam

Academic Editor

PLOS ONE
---

## [Editor Report · Acceptance letter]

PONE-D-24-39287R2

PLOS ONE

Dear Dr. Brovold,

I'm pleased to inform you that your manuscript has been deemed suitable for publication in PLOS ONE. Congratulations! Your manuscript is now being handed over to our production team.

Kind regards,

on behalf of

Dr. Rehana Abdus Salam

Academic Editor

PLOS ONE